# Parameter inference for stochastic biochemical models from perturbation experiments parallelised at the single cell level

**Anđela Davidović**[1], **Remy Chait**[2,3], **Gregory Batt**[1,4], **Jakob Ruess**[1,4]*

**1** Department of Computational Biology, Institut Pasteur, Paris, France, **2** Biosciences, Living Systems Institute, University of Exeter, Exeter, The United Kingdom, **3** Institute of Science and Technology Austria, Klosterneuburg, Austria, **4** Inria Paris, Paris, France

* jakob.ruess@inria.fr

## Abstract

Understanding and characterising biochemical processes inside single cells requires experimental platforms that allow one to perturb and observe the dynamics of such processes as well as computational methods to build and parameterise models from the collected data. Recent progress with experimental platforms and optogenetics has made it possible to expose each cell in an experiment to an individualised input and automatically record cellular responses over days with fine time resolution. However, methods to infer parameters of stochastic kinetic models from single-cell longitudinal data have generally been developed under the assumption that experimental data is sparse and that responses of cells to at most a few different input perturbations can be observed. Here, we investigate and compare different approaches for calculating parameter likelihoods of single-cell longitudinal data based on approximations of the chemical master equation (CME) with a particular focus on coupling the linear noise approximation (LNA) or moment closure methods to a Kalman filter. We show that, as long as cells are measured sufficiently frequently, coupling the LNA to a Kalman filter allows one to accurately approximate likelihoods and to infer model parameters from data even in cases where the LNA provides poor approximations of the CME. Furthermore, the computational cost of filtering-based iterative likelihood evaluation scales advantageously in the number of measurement times and different input perturbations and is thus ideally suited for data obtained from modern experimental platforms. To demonstrate the practical usefulness of these results, we perform an experiment in which single cells, equipped with an optogenetic gene expression system, are exposed to various different light-input sequences and measured at several hundred time points and use parameter inference based on iterative likelihood evaluation to parameterise a stochastic model of the system.

**Data Availability Statement:** All data and code used is publicly available on a Gitlab repository at https://gitlab.pasteur.fr/adavidov/inferencelnakf. We have also used Zenodo to assign a DOI to the

repository: https://doi.org/10.5281/zenodo.5229416.

**Funding:** JR was supported by the ANR grant CyberCircuits (ANR-18-CE91-0002). The funders had no role in study design, data collection and analysis, decision to publish, or preparation of the manuscript.

**Competing interests:** The authors have declared that no competing interests exist.

## Author summary

A common result for the modelling of cellular processes is that available data is not sufficiently rich to uniquely determine the biological mechanism or even just to ensure identifiability of parameters of a given model. Perturbing cellular processes with informative input stimuli and measuring dynamical responses may alleviate this problem. With the development of novel experimental platforms, we are now in a position to parallelise such perturbation experiments at the single cell level. This raises a plethora of new questions. Is it more informative to diversify input perturbations but to observe only few cells for each input or should we rather ensure that many cells are observed for only few inputs? How can we calculate likelihoods and infer parameters of stochastic kinetic models from data sets in which each cell receives a different input perturbation? How does the computational efficiency of parameter inference methods scale with the number of inputs and the number of measurement times? Are there approaches that are particularly well-suited for such data sets? In this paper, we investigate these questions using the CcaS/CcaR optogenetic system driving the expression of a fluorescent reporter protein as primary case study.

## 1 Introduction

Finding appropriate mathematical models to represent biological processes inside cells is one of the core challenges of computational biology. In particular, the last decade has seen much work focused on the development of methods to infer parameters of mechanistic models of biochemical reaction networks from experimental data [1–4]. A common premise in most of these works is that models are complex and unknown parameters numerous, with available data being sparse and noisy [5]. While the complexity of biological systems has not changed, the availability and reliability of data certainly has. For instance, experimental techniques and platforms nowadays allow us to observe the onset of transcription at single molecule precision, to fully automatically measure the expression levels of genes every couple of minutes, to perturb and drive gene expression in populations or individual cells [6–10], and even to let computational models interact with single cell gene expression processes in real time [11]. To which degree these new capacities will eventually allow us to resolve the ill-posedness of reverse engineering models of biological systems from experimental data remains to be clarified, but in any case the availability of new types of data calls for methods that are capable of exploiting the data in its entirety.

In this paper, we focus on data obtained from microscopy platforms such as the ones presented in [8, 9, 11]. The key feature of these platforms is that microscopy-based observation of gene expression dynamics in single cells is coupled to an optical system (such as a digital micromirror device) that allows the user to target light signals at individual cells. These signals are freely designable and can be time-varying. Cells are in turn equipped with an optogenetic system driving the biological process of interest. In summary, the platforms allow one to perturb the biological process dynamically and in a different way in every cell and to observe possibly very different behaviours of the system of interest at the same time. In other words, these new platforms allow us to parallelise a large number of classical perturbation experiments within a single experiment.

It has been demonstrated in the past that biochemical reactions inside single cells are inherently stochastic [12, 13]. Faithfully capturing single cell microscopy data therefore requires stochastic kinetic models governed by the chemical master equation (CME) [14]. However,

inferring parameters of such models from available data is a formidable task [15, 16]. Solving the CME analytically is rarely possible and numerically approximating its solution based on approaches such as finite state projection [17] or stochastic simulation [18] is computationally prohibitively expensive in most cases. This problem is rendered all the worse if each cell in the experiment is perturbed with a different input stimulation, such that the CME needs to be re-solved for every cell in the experiment. Efficient approximations of the CME, such as the linear noise approximation (LNA) [19], may alleviate the computational burden. However, they may become imprecise if cells are observed over longer time horizons as well as impractical to use when the number of measurement times becomes large and the dimensionality of the data increases.

A more feasible alternative is then to evaluate likelihoods iteratively by exploiting the fact that full likelihoods of single-cell trajectories can be factorized into transition probabilities between subsequent measurement time points. Numerous approaches that exploit such a factorization have been proposed in the past based on various different approximation techniques for the stochastic model, often coupled to sampling schemes for the approximation of posterior distributions over parameter space [20–29]. In [30] a concise explanation of the mathematical basis of such approaches has been provided alongside a brief review of specific techniques that have been used in the past. A key question for the computational efficiency of these methods is whether or not marginal likelihoods that are needed to carry out Bayesian updates of state distributions at measurement time points can be calculated analytically. This is most notably the case when technical measurement errors are normally distributed and the LNA is used as method to approximate the stochastic model (as, for instance, in [22]) since the LNA creates a Gaussian approximation of the state distribution, which constitutes a state prior at measurement times that is conjugate for a Gaussian measurement error model. Alternatively, it is also possible to use techniques other than the LNA to approximate transition probabilities coupled to a Gaussian approximation at measurement times. While computationally convenient, Gaussian approximations are known to be notoriously inaccurate for many systems and it is unclear to which degree this impairs the calculation of single-cell likelihoods in an iterative scheme for the novel types of data that are available these days. In this paper, we focus on coupling general moment closure methods for approximately calculating lower order moments of the CME [31] to Gaussian approximations and analytical Bayesian updates at measurement times to approximate likelihoods of single cell data.

We show that the frequency at which measurements are taken determines the quality of the likelihood approximation. We argue that the approach is ideally suited for data obtained from parallelised single cell experiments in which the collection of measurements is automatically performed every couple of minutes. We find that, if the system is observed with sufficient time resolution, iterative evaluation of likelihoods allows for quite precise inference of parameters even if the system is highly non-linear and the used approximation of the CME is very imprecise over longer time-horizons (for instance if the LNA is applied to a model of a genetic toggle switch that displays bimodal distributions).

Finally, to highlight the practical usefulness of the approach, we perform single-cell experiments to characterise a light-inducible gene expression system. Using simulated data, we show that iterative likelihood evaluation leads to accurate parameter estimation and orders of magnitude less computation time compared to an open loop use of the LNA [19]. This efficiency allows us to perform Bayesian parameter inference for various data sets pooled together from single cells perturbed with different light stimuli and to study how much information about model parameters can be gained by parallelising perturbations in single cells. Using experimental data, we show that parameter estimation can also be effective but might require a proper treatment of non-identifiability.

## 2 Methodology

### 2.1 Stochastic biochemical reaction networks

Consider a reaction network of $S$ chemical species $X_1, \ldots, X_S$ that interact stochastically according to $R$ reactions

$$\alpha_{1r}X_1 + \ldots + \alpha_{Sr}X_S \xrightarrow{a_r(\boldsymbol{X},\boldsymbol{\theta})} \beta_{1r}X_1 + \ldots + \beta_{Sr}X_S, \quad r = 1, \ldots, R, \tag{1}$$

parameterised by $\boldsymbol{\theta} = (\theta_1, \theta_2, \ldots, \theta_P)$ and where the coefficients $\alpha_{sr}$ and $\beta_{sr}$ determine how many molecules of the $s$-th species are consumed and produced in the $r$-th reaction, respectively. If system dynamics are influenced by an input perturbation, $u(t)$, the propensity functions $a_r(\boldsymbol{X}, \boldsymbol{\theta}) = a_r(\boldsymbol{X}, \boldsymbol{\theta}, u(t))$ will additionally depend on this input. Under the assumption that the system is well-stirred and in thermal equilibrium, the probability distribution describing the time evolution of the number of molecules of the different species is governed by the chemical master equation (CME) [14]:

$$\dot{p}(\boldsymbol{x}, t) = -p(\boldsymbol{x}, t)\sum_{r=1}^{R} a_r(\boldsymbol{x}, \boldsymbol{\theta}) + \sum_{r=1}^{R} a_r(\boldsymbol{x} - \boldsymbol{v}_r, \boldsymbol{\theta})p(\boldsymbol{x} - \boldsymbol{v}_r, t), \tag{2}$$

where $\boldsymbol{x} = (x_1, x_2, \ldots, x_S)$ in $\mathbb{N}^S$ is a possible state of the stochastic process $\boldsymbol{X}(t) = (X_1(t), X_2(t), \ldots, X_S(t))$ that counts the numbers of molecules of all species, $p(\boldsymbol{x}, t) := \mathbb{P}(\boldsymbol{X}(t) = \boldsymbol{x})$, and $\boldsymbol{v}_r = \boldsymbol{\beta}_r - \boldsymbol{\alpha}_r$, where $\boldsymbol{\alpha}_r = (\alpha_{1r}, \alpha_{2r}, \ldots, \alpha_{Sr})$, and $\boldsymbol{\beta}_r = (\beta_{1r}, \beta_{2r}, \ldots, \beta_{Sr})$. Since the CME is difficult to solve in most cases, a widely used approach is to derive moment equations from it. However, except if the propensity functions $a_r(\boldsymbol{x}, \boldsymbol{\theta})$ of all reactions are linear in $\boldsymbol{x}$, the time evolution of moments up to any order depends on moments of higher order and the moment equations cannot be solved exactly and need to be approximated using moment closure methods (see Section A in S1 Text).

### 2.2 Data likelihoods and their open loop approximation

Here, we consider microscopy data, $\boldsymbol{y}^{\text{all}}$, that contains information from $N$ cells. The observed data for a single cell is given by $\boldsymbol{y} = \{y_i \mid i = 1, \ldots, M\}$ and described by the measurement model

$$y_i = \boldsymbol{C}\boldsymbol{X}(t_i) + \xi_i, \quad i = 1, \ldots, M, \tag{3}$$

where the $\xi_i$ are independent technical measurement errors assumed to be Gaussian $\xi_i \sim \mathcal{N}(0, \sigma^2)$, the matrix $\boldsymbol{C}$ maps the full system state to the measured output species, and distances between measurement times, $t_{i+1} - t_i = t_m, i = 1, \ldots, M - 1$, are assumed to be all equal for the sake of simplicity.

If cells do not physically interact, the full data likelihood of the data $\boldsymbol{y}^{\text{all}}$ is then the product of single cell likelihoods, $p(\boldsymbol{y}^{\text{all}} | u, \boldsymbol{\theta}) = \prod_{\boldsymbol{y} \in \boldsymbol{y}^{\text{all}}} p(\boldsymbol{y} | u, \boldsymbol{\theta})$, where

$$p(\boldsymbol{y} \,|\, u, \boldsymbol{\theta}) = p(y_1, \ldots, y_M \,|\, u, \boldsymbol{\theta}), \tag{4}$$

and $u = u(t)$ is the time-varying input perturbation to which the cell has been exposed. The $M$-dimensional single cell likelihoods are determined by the distribution of the technical errors $\xi_i$, as well as the joint distribution of the system state over all measurement time points,

$$p(\boldsymbol{x} \,|\, u, \boldsymbol{\theta}) = p(\boldsymbol{x}_1, \ldots, \boldsymbol{x}_M \,|\, u, \boldsymbol{\theta}) := \mathbb{P}(\boldsymbol{X}(t_1) = \boldsymbol{x}_1, \ldots, \boldsymbol{X}(t_M) = \boldsymbol{x}_M \,|\, u, \boldsymbol{\theta}), \tag{5}$$

where $\boldsymbol{x} = \{\boldsymbol{x}_i \mid i = 1, \ldots, M\}$ and $\boldsymbol{x}_i = (x_{1i}, x_{2i}, \ldots, x_{Si})$ are possible full states of the reaction network at times $t_i, i = 1, \ldots, M$. It is important to point out that for microscopy data where individual cells are followed in time, the likelihood does not factorise over time points as would be

the case for other data types such as flow cytometry data [15]. The fundamental difficulty of any likelihood-based parameter inference scheme is that evaluating (4) for given parameters, $\boldsymbol{\theta}$, and given input, $u$, requires one to calculate $p(\boldsymbol{x} \mid u, \boldsymbol{\theta})$ in (5), which is only possible when the CME is tractable.

An approach, termed *open loop LNA-based inference* in the following, that has experienced quite some popularity in the past has therefore been to replace the exact likelihood in (4) by an approximation derived from the linear noise approximation [19, 32–34]. The linear noise approximation, contrary to typical moment closure methods, does not only provide approximations of moments at given time points $t$, but also automatically approximates any inter-time distributions by Gaussian distributions. These distributions can readily be calculated from the model. In sum, this approach enables evaluation of (5) (and (4)) through an approximation (see [19] for details)

$$p(\boldsymbol{x} \mid u, \boldsymbol{\theta}) \approx \mathcal{N}(\boldsymbol{\mu}(u, \boldsymbol{\theta}), \boldsymbol{\Sigma}(u, \boldsymbol{\theta})). \tag{6}$$

While the LNA as such is computationally very efficient, it is important to point out that likelihood evaluations may still become very costly for parallelised single cell experiments: if cells can be tracked and measured automatically over very long time horizons (as in [11]), the number of measurement time points, $M$, can range in the order of several hundreds, implying that the distributions in (6) are extremely high dimensional and that the calculation of the covariance matrix, $\boldsymbol{\Sigma}(u, \boldsymbol{\theta})$, requires a significant computational effort despite the otherwise efficient LNA. This problem becomes ever more severe considering that $\boldsymbol{\Sigma}(u, \boldsymbol{\theta})$ needs to be recalculated for every different input perturbation, $u$, that is used in an experiment and for every point, $\boldsymbol{\theta}$, in parameter space that is explored during parameter inference. In addition, it is long known that the approximation of $p(\boldsymbol{x} \mid u, \boldsymbol{\theta})$ by a Gaussian distribution is very inaccurate or even completely useless in many cases because the system dynamics are highly non-linear and the probability distribution of the system is far from Gaussian. Typically, the LNA should not be used for systems such as genetic toggle switches since the approximation of the real bimodal system distributions with Gaussian distributions is unacceptably imprecise. One aim of this paper is to show that the LNA, and other moment approximation methods, can nevertheless sometimes be used for parameter inference of systems like genetic toggle switches.

## 2.3 Approximating likelihoods using moment equations with filtering

An alternative way of calculating the likelihood can be obtained if we rewrite the joint probability distribution in (4) in terms of transition probabilities between measurement time points as

$$p(y_1, \ldots, y_M) = p(y_1) \cdot p(y_2 \mid y_1) \cdots p(y_M \mid y_{M-1}, \ldots, y_1), \tag{7}$$

where we have omitted the dependence on $u$ and $\boldsymbol{\theta}$ for the sake of readability. The transition probabilities between data points appearing on the right hand side can in turn be rewritten in terms of the technical noise distribution, transition probabilities between full states of the system, and posterior distributions over the system state given past observations:

$$p(y_i \mid y_{i-1}, \ldots y_1) = \int \int \underbrace{p(y_i \mid \boldsymbol{x}_i)}_{\text{tech. noise}} \cdot \underbrace{p(\boldsymbol{x}_i \mid \boldsymbol{x}_{i-1})}_{\text{trans. prob.}} \cdot \underbrace{p(\boldsymbol{x}_{i-1} \mid y_{i-1}, \ldots, y_1)}_{\text{state posterior}} d\boldsymbol{x}_i d\boldsymbol{x}_{i-1}. \tag{8}$$

While exactly evaluating (8) is clearly difficult and requires the solution of the CME due to the appearance of the transition probabilities $p(\boldsymbol{x}_i \mid \boldsymbol{x}_{i-1})$, it is important to note that reasonable approximations can be obtained under assumptions that are much weaker and much more

often fulfilled than the classically used assumption in (6). In particular, if we borrow from the LNA nothing but the assumption that transition probabilities $p(\boldsymbol{x}_i \mid \boldsymbol{x}_{i-1})$ between subsequent measurement times are Gaussian and use some moment approximation method to calculate their means and covariance matrices, (8) can readily be evaluated without requiring the full $p(\boldsymbol{x} \mid u, \boldsymbol{\theta})$ in (6) to be anything close to Gaussian.

Concretely, (8) can be evaluated as follows using an iterative scheme (which is more or less the same as classical Kalman filtering), as graphically presented in Fig 1B.

**Algorithm 1**

*1. Calculate approximate moments up to order two, $\eta^1_{\boldsymbol{x}_1}, \eta^2_{\boldsymbol{x}_1}$, of $p(\boldsymbol{x}_1)$ by moment closure (see Section A in S1 Text).*

*2. Approximate the true $p(\boldsymbol{x}_1)$ by a Gaussian distribution that has $\eta^1_{\boldsymbol{x}_1}, \eta^2_{\boldsymbol{x}_1}$ as moments.*

*3. Given the Gaussian model of the technical measurement errors, this implies that also the marginal likelihood $p(y_1) = \int p(y_1 \mid \boldsymbol{x}_1)p(\boldsymbol{x}_1)d\boldsymbol{x}_1$ is Gaussian and it can be calculated from $\eta^1_{\boldsymbol{x}_1}, \eta^2_{\boldsymbol{x}_1}$ and $\sigma$. Evaluate $p(y_1)$ and store it for the likelihood calculation in (7).*

*4. Since the state prior $p(\boldsymbol{x}_1)$ and likelihood $p(y_1 \mid \boldsymbol{x}_1)$ are Gaussian, the state posterior $p(\boldsymbol{x}_1 \mid y_1)$ is also a Gaussian distribution that can be calculated from $p(\boldsymbol{x}_1)$ and $p(y_1 \mid \boldsymbol{x}_1)$ thanks to Bayes' theorem, as is classically done in Kalman filtering.*

*5. Extract the moments up to order two, $\eta^1_{\boldsymbol{x}_1|y_1}, \eta^2_{\boldsymbol{x}_1|y_1}$, of $p(\boldsymbol{x}_1 \mid y_1)$.*

*6. Solve moment equations (see Section A in S1 Text) over $t_m$ time units (i.e. over $[t_1, t_2]$) using $\eta^1_{\boldsymbol{x}_1|y_1}, \eta^2_{\boldsymbol{x}_1|y_1}$ as initial conditions in order to obtain moments $\eta^1_{\boldsymbol{x}_2|y_1}, \eta^2_{\boldsymbol{x}_2|y_1}$ that approximate the moments of the distribution $p(\boldsymbol{x}_2 \mid y_1)$.*

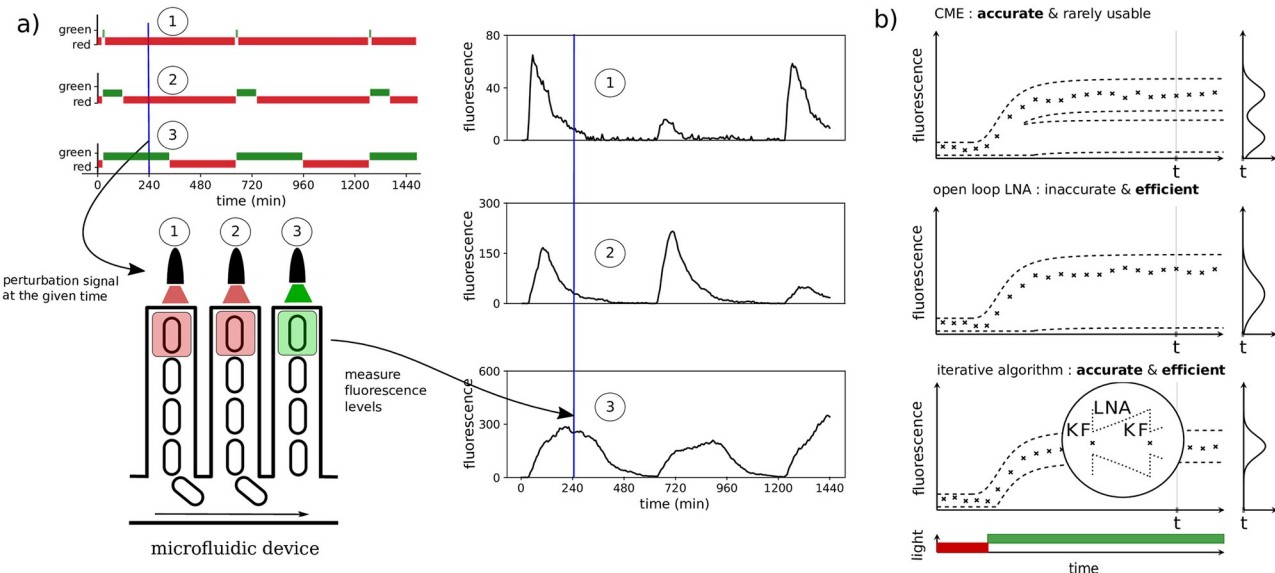

**Fig 1. a) Schematic representation of experiments parallelised at the cell level**. Top left: an example of different light patterns that drive protein production in respective cells. Green light is used to excite the light system in a single cell and to start the production of a fluorescent protein, and red light is used to stop production. Bottom left: representation of a microfluidic device called a mother machine: each microchannel is of the width of a single cell. Cells are kept in a constant environment and are growing and dividing. One cell stays at the bottom of a channel, while the others are being washed away. Each channel is used to obtain longitudinal data at the single cell level. Each channel is then given a light pattern that can be pre-defined (as in top left) or adjusted in real-time in response to incoming data. For more details see [11]. Right: fluorescence data corresponding to the light stimulation profiles at top left. **b) Benefit of coupling the LNA to Kalman filtering for parameter inference of biochemical reactions**. Top: The chemical master equation can be used to calculate exact likelihoods but is rarely solvable in practice. An example situation for a system that exhibits bimodality is shown (crosses: data, dashed lines: the CME solution is visualised via confidence regions). Middle: The LNA provides efficient approximations but is based on often very inaccurate Gaussian approximations of the full CME solution. Bottom: Coupling the LNA with Kalman filtering (Algorithm 1) requires only transition probabilities over short time intervals to be close to Gaussian.

*7 Iterate*: $p(\boldsymbol{x}_2 \mid y_1)$ *is approximated by a Gaussian equivalently to* $p(\boldsymbol{x}_1)$ *in the step 2. and so forth.*

At the end of computation, we obtain an approximate scheme for calculating the likelihood that only requires transition distributions between subsequent measurement times to be sufficiently close to Gaussian, and that "corrects" the *open loop* approximation in moment equations by re-conditioning on the data at every measurement time.

**Remark 1** *It is important to understand that conditioning on the data to evaluate likelihoods leads to an approximation whose accuracy depends crucially on the data and not only on how the stochastic model is approximated. This means that we need to carefully investigate this dependence, but overall it is an important strength of the approach as it allows us to deal with models for which all existing approximation methods fail if applied in open loop.*

## 3 Results

### 3.1 Accurate likelihood approximations using inaccurate approximations of the chemical master equation

The core advantage of the approach in Algorithm 1 compared to open loop likelihood approximations based directly on Eqs (4) and (5) is that the approximation of the CME only needs to be accurate over the time that passes in between subsequent measurement steps, which, in many cases, is a much less restrictive requirement. For instance, moment closure methods have been found to diverge or return negative variances when solved over longer time horizons. As we mentioned earlier, Gaussian approximations, such as used in the LNA, are necessarily bad for bimodal distributions displayed by some systems. It is important to understand, however, that what is typically meant by "a system displaying bimodal distributions" is that the solution of the CME becomes bimodal when sufficient time passes but this does not necessarily imply that transition distributions between subsequent measurement time points are also bimodal. For instance, when a genetic toggle switch [7] is started in one of its stable equilibrium points, then after a short amount of time the system will still most likely be close by and the solution of the CME may still be unimodal around the equilibrium point. If measurements are taken frequently, Algorithm 1 only requires approximations of such short term transition distributions from given "initial states" in step 6 and may therefore lead to accurate approximations of the likelihood even if the deployed CME approximation deteriorates over longer time horizons.

**3.1.1 Studying accuracy of likelihood approximations with a simple positive feedback loop system.**   To demonstrate accuracy of likelihood approximations, we start by investigating a simple reaction network containing only a single chemical species, X. In this study case, accurate solution of the full chemical master equation using finite state projection (FSP) [17] is straightforward and approximated likelihoods can readily be compared to the true likelihoods (see Section B in S1 Text). To nevertheless obtain an example that is sufficiently interesting and challenging, we assume that the production of X is a non-linear function of the abundance of X such that a positive feedback loop is formed and the system displays bimodal distributions (see Fig 2A). Concretely, we assume that production and degradation of X occur according to

$$\emptyset \xrightarrow{a_0 + a \cdot h(X)} \mathrm{X}, \qquad \mathrm{X} \xrightarrow{d} \emptyset, \qquad (9)$$

where $h(x) = \frac{x^n}{x^n + k^n}$ and any possible dependence on input perturbations, $u(t)$, is, for now, omitted. We use the linear noise approximation not only because it is bound to be inaccurate for this system, but also because most moment closure methods are easy to use only for mass-

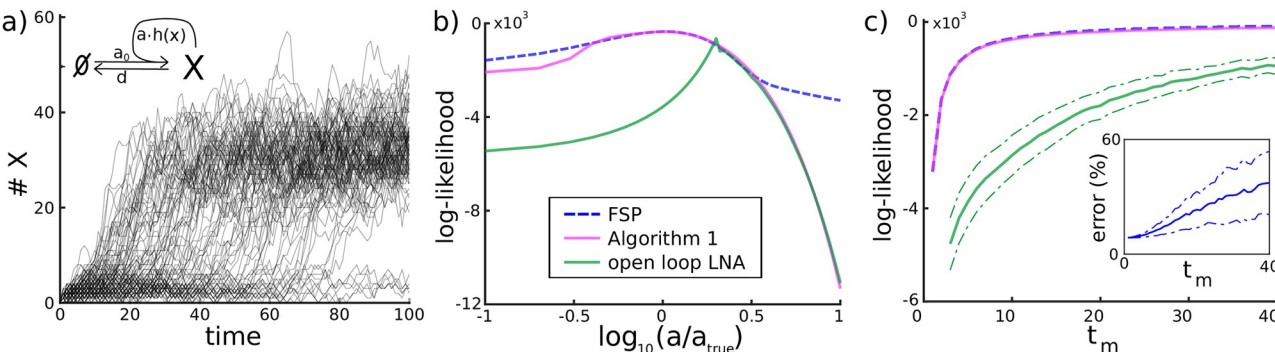

**Fig 2. Accuracy of the likelihood approximation.** A toy example of a reaction network (9) with a single chemical species and a positive feedback loop. **a)** $N = 100$ system trajectories obtained using Gillespie's stochastic simulation algorithm. Molecule numbers of X are either very low, because only basal production of X is active at rate $a_0$, or they switch to higher levels where the positive feedback is active. **b)** Data likelihood $p(y \mid \theta)$ for a single cell for the distance between measurement times $t_m = 10$ as a function of $\theta = a$ calculated using different approaches. Blue dashed: true data likelihood up to very small errors obtained by FSP with a large truncation set. Purple: approximation using Algorithm 1. Green: approximation based on open loop use of the LNA. **c)** Mean ± one standard deviation of data likelihood $p(y_{i,t_m} \mid \theta)$ averaged over $N = 100$ cells as a function of the distance between measurement times $t_m$. The colour coding is the same as in panel b. *Inset*: Relative error of Algorithm 1 as a function of $t_m$. Parameter values and initial condition for this system have been chosen as $a_0 = 2$, $a = 8$, $k = 20$, $n = 5$, $d = 0.25$, $\sigma = 4$, $X(0) = 0$, $t_S = 100$, all in arbitrary or no units.

action kinetics and require further adjustments when the propensities are given as Hill functions.

To investigate the accuracy of Algorithm 1, we assume that the system can be observed every $t_m$ time units for various values of $t_m$ and that data is collected according to the measurement model given in (3). For the sake of an easily understandable presentation, we decided to study the likelihood, $p(y \mid \theta)$, only for a single cell and only as a function of a single parameter, $a$, while the remaining parameters are fixed. Fig 2B shows that, for $t_m = 10$, the true likelihood and its approximation according to Algorithm 1 agree in the relevant parameter regime and have almost exactly the same maximum, implying that the correct maximum-likelihood estimator would be obtained with the approximated likelihood.

To compare this result to the open loop approach, we also evaluated the likelihood using the LNA and (6). Despite the fact that the same approximation of the CME is used for the same system, the approximated likelihoods are very different and the open loop approach is very inaccurate almost everywhere in parameter space. This is not surprising since the bimodal distributions displayed by the system are ill approximated by Gaussian distributions.

On the other hand, since measurements are taken more frequently than the time that the system needs to transition to larger molecule counts, the transition distributions between subsequent measurements are sufficiently close to Gaussian to render the LNA useful when it is deployed in Algorithm 1. The exception to this is when the value of the parameter $a$ is very large such that proteins are produced very quickly once the positive feedback loop is triggered. In this case, transition distributions between measurement time points are also bimodal and the likelihood approximation according to Algorithm 1 yields the same inaccurate result as the open loop approach. However, since the true value of the parameter considered in this case study is not that large, this region of parameter space is not of particular importance for the purpose of parameter inference.

To further benchmark Algorithm 1, we calculated the likelihood at the true parameters for $N = 100$ cells as function of $t_m$. Fig 2C shows the average likelihood at the true value of the parameter $a$ as a function of $t_m$ as well as standard deviations around it. It can be seen that open loop application of the LNA leads to extremely imprecise approximations for almost all

$t_m$, while Algorithm 1 is very accurate except when $t_m$ is large and measurements are too distant from each other for transition distributions to be close to Gaussian (see the relative error in the inset).

**3.1.2 Using the LNA iteratively to infer parameters of a toggle switch.** Genetic toggle switches are systems consisting of two proteins that repress each others production. Since such systems are typically constructed to implement bistability and switching behaviour, they are typically considered as the role model of systems for which one should not use the LNA. However, our results in the previous section suggest that it might in fact be possible to accurately infer parameters of a toggle switch using the LNA in Algorithm 1. To test this, we consider the following model of a genetic toggle switch:

$$\emptyset \xrightarrow{a_{A0}+h_A(B)} A \qquad A \xrightarrow{d_A} \emptyset$$
$$\emptyset \xrightarrow{a_{B0}+h_B(A)} B \qquad B \xrightarrow{d_B} \emptyset \qquad (10)$$

where $h_A(x) = \frac{a_A}{x^{n_A}+k_A}$ and $h_B(x) = \frac{a_B}{x^{n_B}+k_B}$ are Hill functions modelling the repression of the production of protein A by protein B and of B by A, respectively. The resulting system trajectories switch stochastically between two regimes where either A is present at high copy numbers and the gene that produces protein B is repressed or B is present at high copy numbers and the gene that produces protein A is repressed (Fig 3A). We assume that only one of the proteins, B, can be measured every $t_m$ time units in $N = 10$ cells with measurement noise as in (3).

To test if it is in principle possible to use the LNA for parameter inference for this system without confounding the results by identifiability problems, we assumed that only a single parameter, namely $d_B$, is unknown and used a Metropolis Hastings Markov chain Monte Carlo (MCMC) method based on log-normal proposal distributions with fixed variance for Bayesian inference, assuming a flat prior distribution for $d_B$. We find that the MCMC algorithm generally converges very quickly to the vicinity of the true value of $d_B$ (Fig 3B). However, when the time between measurements is comparably large ($t_m = 20$) the algorithm fluctuates around a value that is slightly smaller than the true value. Since the data is sufficiently rich to, in principle, allow for a very precise estimation of only a single parameter, we can attribute this error to approximation errors in the likelihood.

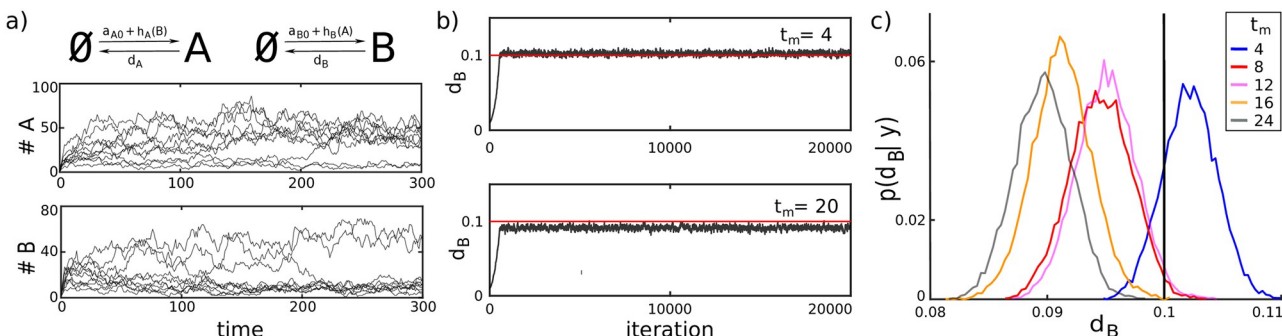

**Fig 3. Parameter inference for a genetic toggle switch using the LNA in Algorithm 1. a)** $N = 10$ system trajectories obtained using Gillespie's stochastic simulation algorithm. **b)** The MCMC search (20000 steps) with data from $N = 10$ cells measured every $t_m = 4$ (top) and $t_m = 20$ (bottom) time units shows fast convergence from an initial guess of $d_B = 0.01$ to an approximated posterior distribution whose maximum is close to, but not exactly at, the true parameter value. The error is larger for the case where $t_m = 20$. **c)** Posterior distributions for different values of $t_m$: $t_m = 4$ (blue), $t_m = 8$ (red), $t_m = 12$ (purple), $t_m = 16$ (orange), $t_m = 20$ (grey). All distributions were obtained as histograms of the values visited by the corresponding Markov chain with a burn in period of 1000 steps. The magnitude of the error of the MAP estimator tends to become larger with increasing $t_m$ as expected from the increasing error of the employed likelihood approximation. Parameters for this case study have been chosen as $a_{A0} = 0.5$, $a_{B0} = 0.5$, $a_A = a_B = 700$, $k_A = k_B = 100$, $n_A = n_B = 2$, and $d_A = d_B = 0.1$, $A(0) = B(0) = 1$, $\sigma = 4$, $t_S = 300$ all in arbitrary or no units. The model is given in (10).

To investigate in more detail how this error depends on the distance between measurement times, $t_m$, we repeated the parameter search for $t_m \in \{4, 8, 12, 16, 20\}$ and found that the error of the maximum a posteriori (MAP) estimator increases with $t_m$ but remains around 10% even for the largest considered $t_m = 20$. This implies that despite the fact that the likelihood approximation becomes increasingly imprecise for large $t_m$, the maximum of the corresponding posterior distribution still remains at values that are reasonably close to the true parameter value. We therefore conclude that the errors in the estimation of $d_B$ obtained here are overall very small, in particular for $t_m = 4$. This is quite remarkable considering that, to the best of our knowledge, no other successful use of any moment-based parameter estimation scheme has been reported up to date for systems like a genetic toggle switch. That said, it should be noted that the model considered here switches relatively slowly between the two regions. For faster switching systems more frequent measurements would be required for a LNA-based likelihood approximation to remain accurate.

## 3.2 Bayesian inference for experiments parallelised at the single cell level

Having demonstrated that the likelihood approximation can be accurate and used to infer parameters even when the system dynamics are complex, we return to the main objective of this paper: to investigate scalability of the approach and to test if it is practically usable for experiments parallelised at the cell scale, where each cell is perturbed with a different input.

To this end, we focus on the stochastic model that was used in [11] to describe single cell responses to light of the CcaS/CcaR optogenetic system. Upon exposure to green light, CcaS flips to an active state and phosphorylates the response regulator CcaR, which then activates the production of a fluorescent protein [35]. Red light reverts this process and stops gene expression very quickly.

In [11], the following single cell gene expression model for the CcaS/CcaR optogenetic system was used:

$$\emptyset \xrightarrow{h_0} E \xrightarrow{h} \emptyset, \quad \emptyset \xrightarrow{aE(t)L(t)} F \xrightarrow{b} \emptyset, \tag{11}$$

where $E$ is a generic variable that was named "cell responsiveness" and which pools together sources of variability that are extrinsic to the studied gene (e.g. variations in the number of ribosomes or plasmid copy number fluctuations). The variable $E$ can therefore be thought of as a model of extrinsic noise that differs from classical extrinsic noise models [15, 36, 37] in that it is time-varying. Allowing extrinsic noise factors to fluctuate is necessary for our data due to long duration of our experiments (several tens of cell generations). Notably, this implies that the model contains a parameter, $h$, that characterizes the time scale of extrinsic noise fluctuations, and that can be inferred from experimental data. The second random variable, $F$, quantifies the amount of fluorescent protein present in a cell and is directly measured in experiments (up to a scaling factor $s$, see Section C in S1 Text for details). The single cell protein production rate is determined by the cell responsiveness, $E(t)$, together with the non-linear time-dependent variable, $L(t)$, which is indirectly controlled by the external light signal, $u(t)$. More precisely,

$$L(t) = \frac{(c \cdot l(t))^n}{(c \cdot l(t))^n + k^n}, \tag{12}$$

where $l(t)$ evolves according to

$$\frac{dl(t)}{dt} = u(t) - c \cdot l(t), \tag{13}$$

where the light signal $u(t) = 1$ if the cell is exposed to green light at time $t$, and $u(t) = 0$ if the cell is exposed to red light. Overall, the model contains 8 parameters that need to be inferred from the data, $\boldsymbol{\theta} = \{a, b, s, m, h, c, n, k\}$, where $m := \frac{h_0}{h}$ is the mean of the stationary Poisson distribution determined by the dynamics of $E$.

**3.2.1 Inference of parameters from simulated data.** To test if parameters can be better learned from parallelised experiments at the single cell level, we fixed the parameters of the model as in Table C.1 and used Gillespie's stochastic simulation algorithm to simulate data sets in which different numbers of cells are exposed to various light patterns. We then used a Metropolis-Hastings MCMC algorithm to perform Bayesian inference of the model parameters for each of the data sets, in order to test how well the true parameter values can be learned in these cases.

Concretely, we consider experiments in which each cell is exposed to one of 6 different light patterns (see Fig 4). We categorise these 6 light patterns into those with short, intermediate and long green signals. Light pattern 1 has a multiple short green signals that last only 12 min, light patterns 2, 3, 4 and 5 fall into the intermediate category with green signals lasting between 30 min and 300 min, and the light pattern 6 has a long green signal that lasts until the end of experiment. We simulated 10 cells for each of the six light profiles so as to obtain a data set with 60 cells in total. This set of cells forms a mixed group called Group 0 (G0). Then, we

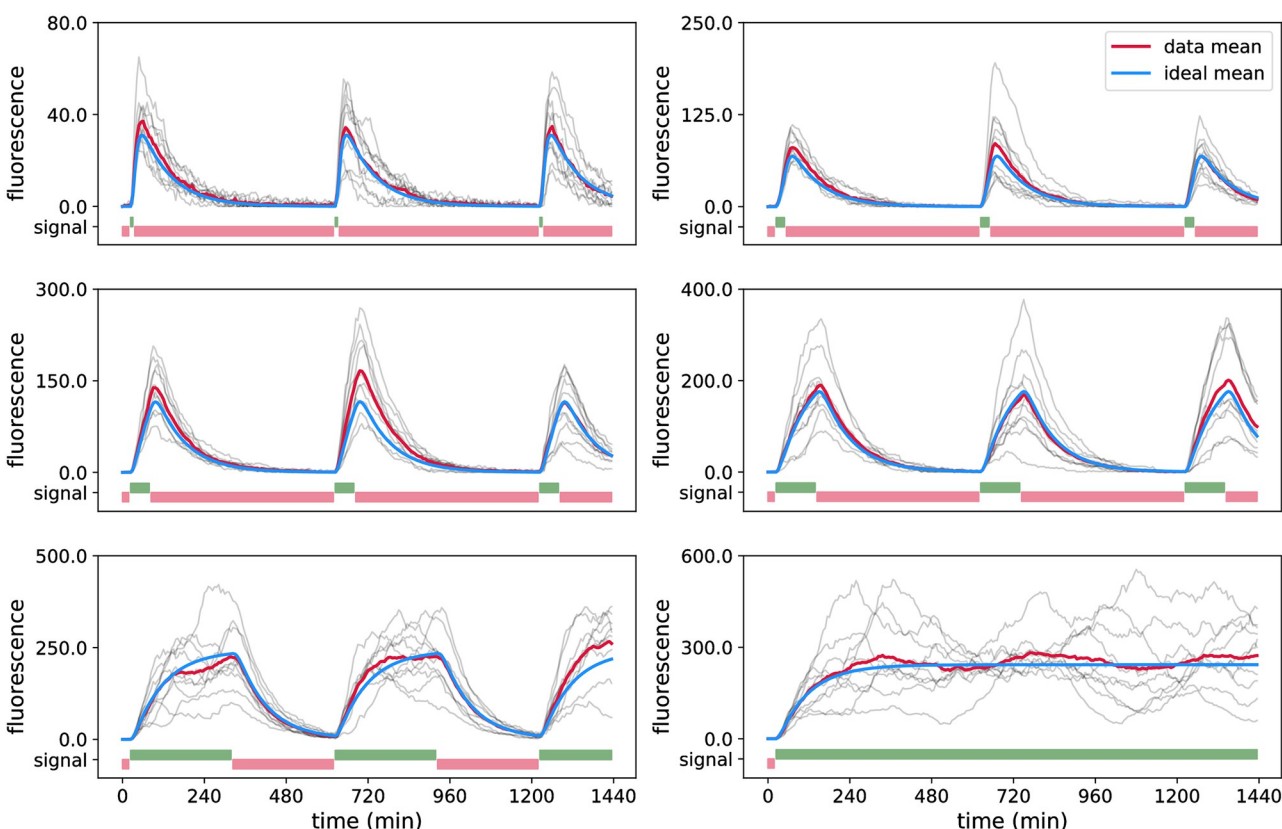

**Fig 4. Simulated data for the CcaS/CcaR optogenetic system.** One data set used for parameter inference for G0 is shown in the six panels. The different panels display cells exposed to different light inputs, $u(t)$. The pulses of red and green light inputs are shown in red and green bars, respectively, at the bottom of each panel. Note that each light pattern starts with red light. Simulated data for 60 cells in total, 10 cells per panel, is shown in grey, the mean of the displayed cells in red, and the true mean (i.e. for infinitely many cells) is shown in blue. Parameter values are provided in Table A in S1 Text. Simulated data sets for other groups and different numbers of cells are shown in Figs A-C in S1 Text.

simulated an additional 60 cells, all exposed to the short light pattern 1, and named it Group 1 (G1). We repeated the same process for two light patterns from the intermediate category, light patterns 2 and 5, and created Group 2 (G2) and Group 5 (G5), and again for the long green signal and created dataset Group 6 (G6). We decided to omit results for light patterns 3 and 4 as these results do not change our conclusions and for the sake of readability of the figures that follow.

As in the actual experimental setup, measurements are taken every $t_m$ = 6 min up to a final time of $t_M$ = 1440 min, implying that there are 240 measurements per cell and that open loop calculation of likelihoods based on approaches such as in (6) would be computationally difficult or unfeasible. On the other hand, likelihood evaluation using Algorithm 1 takes only a fraction of a second per cell. Nevertheless, we need to explore an 8-dimensional parameter space, which remains challenging since some of the parameters affect the measured output similarly and cause identifiability problems.

Running the MCMC algorithm for 10 000 iterations, we find that the quality of parameter estimates depends strongly on the group of cells that is used for inference, meaning that the identifiability of parameters depends on which light input is applied to cells. Diversified light signals in cells (as in G0) lead to relatively tight posterior distributions for all parameters and MAP estimates that are close to the true values of the parameters (see Fig 5).

When all cells receive the same light input (G1, G2, G5, G6) some of the parameters are not identifiable in most cases. In particular, posterior distributions for G1 and G6 are very broad for some parameters. Cells in G6 are exposed permanently to green light and the full dynamics of the system are never exposed and cannot be characterised (parameters $a$, $b$, $s$ and $c$, see Fig 5A). Cells in G1 are exposed to only very short pulses of green light which makes it difficult to reliably learn parameters that characterise fluctuations in the cells' responsiveness $E$ (parameters $a$, $s$, $m$ and $h$, see Fig 5A). This is because $E$ influences the measured output only in the presence of green light, that is when protein production is active. Data from G2 have better posterior distributions than G6 or G1, but still lead to similar identifiability problems as in G1 for certain parameters (parameters $a$ and $s$, see Fig 5A).

However, data from G5 leads to posterior distributions that are of similar quality to the mixed group G0 (see Fig 5A and Fig FA in S1 Text.) implying that the light inputs of this group excite the system dynamics in a sufficiently rich way, at least if 60 cells are used for parameter inference. Using fewer cells would, however, be advantageous. On the one hand because evaluating likelihoods of many cells is computationally costly, and on the other hand because cells in the experiment that are not needed for the calibration of models can be used for other purposes, for instance for validating model predictions.

We therefore investigated how the quality of inferred parameter estimates for the different groups changes when the total number of cells is reduced. We additionally simulated smaller sets of data for the same groups G1, G2, G5 and G6, with the total number of 12 and 30 cells. Similarly, we simulated the corresponding smaller data sets for G0 with 2 and 5 cells per light pattern, respectively (Figs A-C in S1 Text). We find that when cells receive diversified light signals (G0), using only 12 cells (2 each for any of the six light inputs) leads to quite broad posterior distributions but using 30 cells (5 cells per light input) leads to posterior distributions and parameter estimates that are almost as good as when 60 cells are used (Fig 5B). Using 30 cells but applying the same light input to all of them, however, leads to broader posterior distributions (Figs D-K in S1 Text).

We can therefore conclude that exposing different cells to different perturbations within the same experiment provides means to increase the information content in the data and thereby allows us to learn model parameters either more precisely or with fewer measured cells. In the very least, a variation of light inputs applied to cells within a single experiment

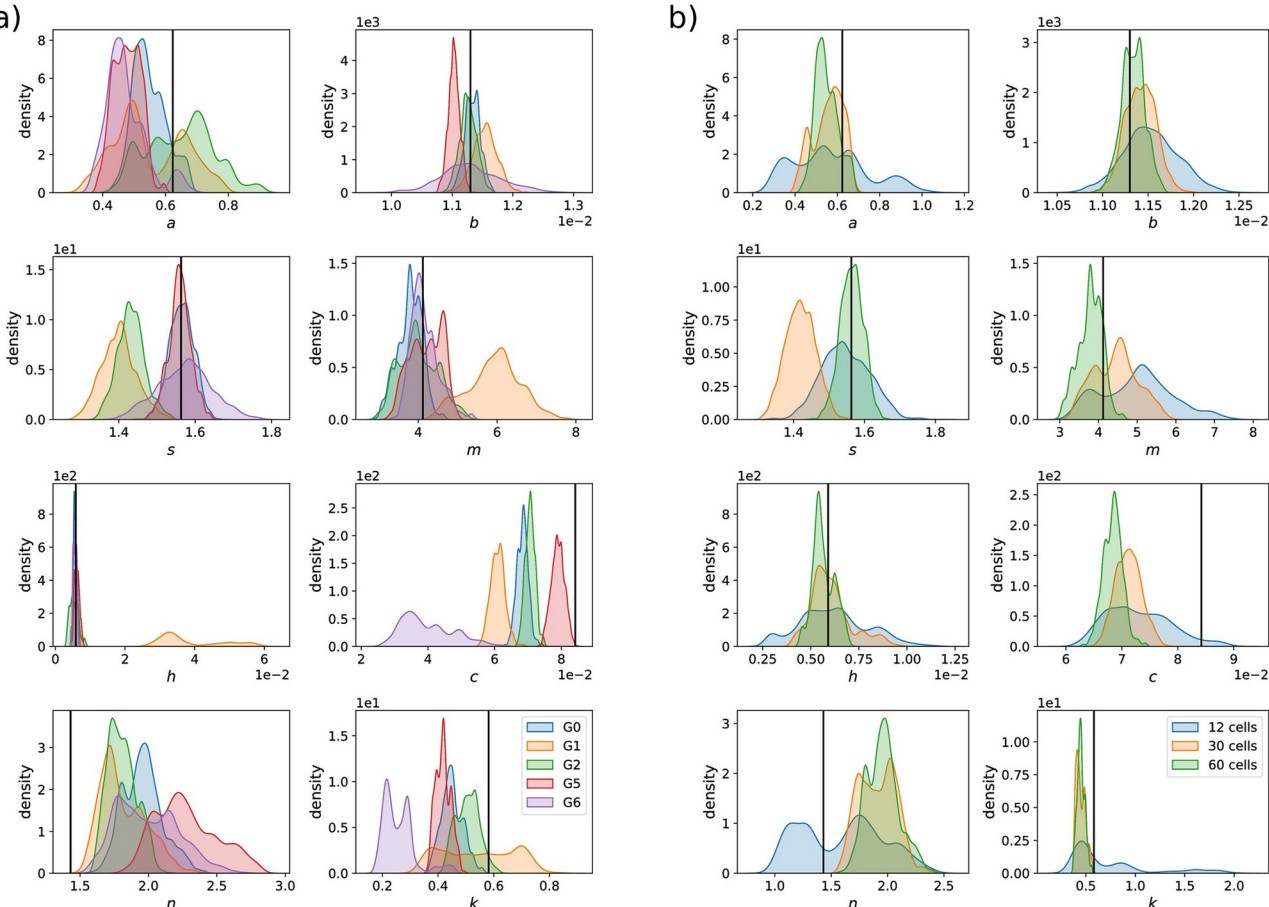

**Fig 5. Inference results for the CcaS/CcaR optogenetic system. a)** One-dimensional marginals of posterior distributions obtained using 60 cells, either all exposed to the same light input (G1—orange, G2—green, G5—red, G6—purple) or to diversified light inputs containing 10 cells for each of the 6 light patterns (G0—blue). The vertical black lines in each panel shows the true value of the corresponding parameter that was used to simulate the data sets. Results for data sets with 12 and 30 cells are provided in Fig D in S1 Text. Two-dimensional marginals showing correlations between model parameters are provided in Figs G-K in S1 Text. **b)** Posterior distributions for the mixed group, G0, for varying numbers of total cells that were used to infer parameters. Results for other groups are provided in Figs E and F in S1 Text. In all cases, posterior distributions have been obtained as histograms of values visited by the MCMC chain of 10 000 iterations, without a burn-in period of 4 000 iterations.

(like in G0) ensures that some of these inputs excite the system dynamics in informative ways and that not all cells are stimulated with light inputs that alone do not provide information on all model parameters (such as in G1, G2 and G6).

**3.2.2 Iterative likelihood evaluation reduces the computational cost by orders of magnitude.** To establish that experiments parallelised at the single cell level are useful for calibrating models, we deployed our likelihood approximation and showed that is it practically applicable for inferring model parameters from data of such experiments. With that established, we focus in more detail on the computational cost of the likelihood evaluation and investigate how it scales with the number of cells, the number of different light inputs that are used, and the number of measurement times for each cell. We stress that it is crucial that the likelihood evaluation is very fast since this calculation needs to be performed at every iteration of the MCMC algorithm, that is at least 10 000 times, to infer parameters in the case studies shown in the previous section.

To be able to compare the computational cost of Algorithm 1 to open loop likelihood calculations, we additionally implemented a method that uses the LNA to directly approximate the full likelihood according to (6). In the open loop approach, once the full probability distribution $p(\boldsymbol{x} \mid u, \boldsymbol{\theta})$ is calculated for given parameters $\boldsymbol{\theta}$, data from all cells that have received the input perturbation $u(t)$ simply need to be plugged into this distribution to calculate the likelihood of the full data set. On the other hand, Algorithm 1 is always operating on the specific data of single cells and needs to be re-run for every cell irrespective of whether or not they all received the same light input. This advantage of the open loop approach naturally becomes less relevant the more diversified the light inputs are that are sent to the cells in the experiment. Furthermore, when the number of measurement times per cell increases, the dimensionality of the data increases and the calculation of $p(\boldsymbol{x} \mid u, \boldsymbol{\theta})$ becomes prohibitively costly, even if the computationally cheap Gaussian approximations of the LNA are used. In contrast to that, if Algorithm 1 is used for the likelihood calculation, additional measurements only require additional iterations of the algorithm's steps, which increases its computational cost only linearly.

In summary, setting aside differences in the accuracy of the likelihood approximations obtained by the different approaches, we expect Algorithm 1 to computationally outperform the open loop approach except if the number of measurements per cell in the experiment is very small. Concretely, we find that if all cells receive the same light input, as in the data considered for G1, Algorithm 1 is computationally superior when the number of measurements is larger than 10 if 6 cells are observed, when the number of measurements is larger than 15 if 18 cells are observed, and when the number of measurements is larger than 20 if 30 cells are observed (Fig 6A). For the data set with 240 measurements per cell that we considered in this paper, the open loop approach would be way too costly to be used at all. When cells are exposed to different light inputs, as in the data considered for G0, Algorithm 1 becomes even more advantageous and computationally outperforms the open loop approach even for 30 cells when only a handful of measurements are taken (Fig 6B).

**3.2.3 Inference of parameters from experimental data.** We established using simulated data that the presented likelihood approximation is computationally efficient and can be used for inference of model parameters when experiments are parallelised at the single cell level. Can we use it in practice?

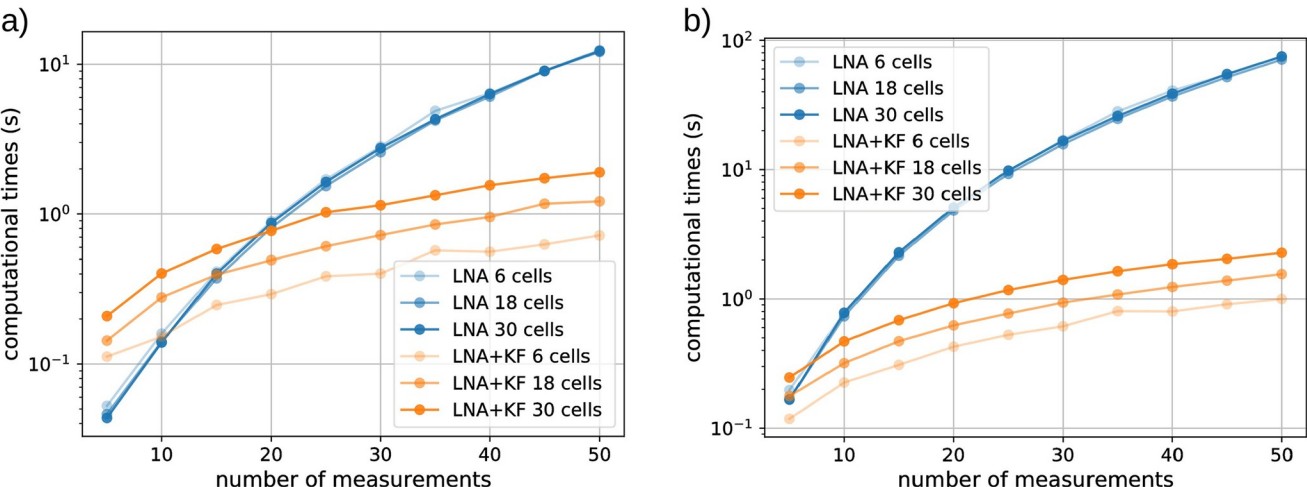

**Fig 6. Evaluation of the computational efficiency of Algorithm 1. a)** The computational cost of likelihood evaluations using Algorithm 1 (orange) for different numbers of cells (differently shaded lines) is compared to open loop likelihood evaluations (blue) when all cells have received the same light input (G1). **b)** Same as panel **a)** except that cells have been exposed to all 6 different light inputs (G0).

We have recently started to use the CcaS/CcaR optogenetic system to drive parts of a repressilator circuit with light. A first practical task is to quantitatively characterise the optogenetic system when plasmids carrying the repressilator circuit are present in cells. To test this, we constructed a strain in which the CcaS/CcaR system is driving a fluorescent reporter protein, the repressilator circuit is present but the CcaS/CcaR system is not coupled to it (see Section D in S1 Text). We performed an experiment with the new strain in which cells are grouped and exposed to light stimulations as in the *in silico* case study in Section 3.2.1. We then manually curated the data and extracted a data set of 30 viable cells (five cells for each of the six groups, see Fig 7) in order to parameterise the model.

Using the Metropolis-Hastings MCMC scheme for parameter inference, we observe that some of the parameters are difficult to identify (Fig O in S1 Text). The reason for this might be the mismatch between the model and real system coupled to unknown, and probably non-Gaussian, experimental measurement noise that makes parameter inference from real data difficult compared to an idealised *in silico* case study. In particular, since the light activation function $L(t)$ is only indirectly connected to the observable fluorescence levels in cells, the multiple parameters that were used in (12) to define its shape are difficult to determine jointly.

We therefore decided to fix 4 of the 8 unknown model parameters to the maximum likelihood estimates extracted from the first run of the Metropolis-Hastings MCMC algorithm (see

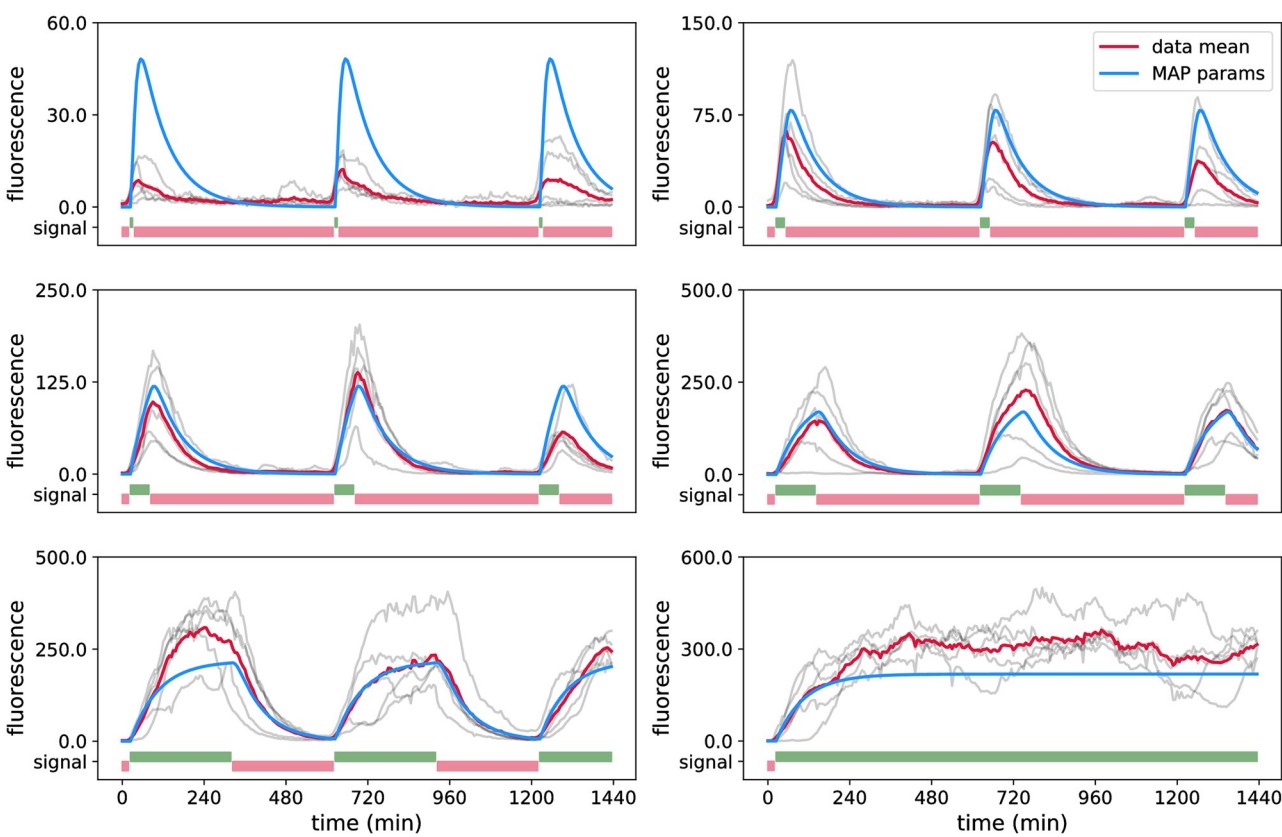

**Fig 7. Experimental data for the CcaS/CcaR optogenetic system.** The curated data set of 30 cells in total, 5 cells per panel, is shown in grey, where each trajectory shows the fluorescence of one cell over a period of 24 hours with the time between subsequent measurement points being 6 min. Each panel displays cells exposed to the same light inputs, $u(t)$. Red and green light inputs are shown in red and green bars, respectively, at the bottom of each panel. Note that each light pattern starts with red light. The mean of the displayed cells is shown in red, and the blue lines are ideal means predicted by the model using MAP estimator obtained from this same data set (see Section C.5 and Table C, MAP (4), in S1 Text).

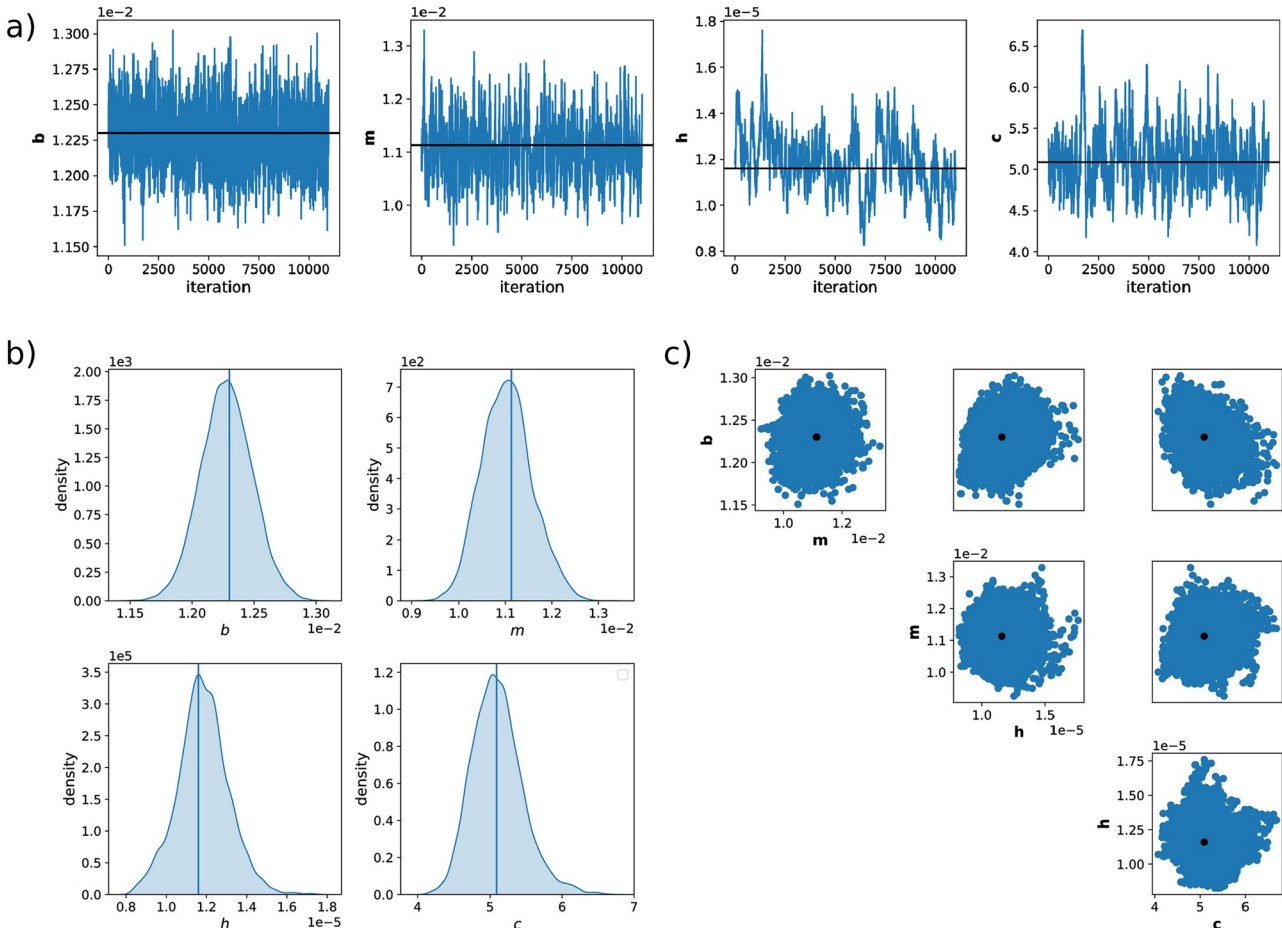

**Fig 8. Parameter inference using Algorithm 1 on experimental data. a)** MCMC chains, after a burn in period, for parameters *b*, *m*, *h* and *c*. Black lines show values and the position in the chain of the MAP estimator. **b)** One-dimensional marginals of posterior distributions. **c)** Two-dimensional posterior marginals of the MCMC parameter search of 14 000 iterations in total and the burn-in period of 3 000 iterations. Black dots show the values of the parameters of the MAP estimator. Data used for inference: 30 cells in total, 5 cells per light group, as in G0, shown in Fig 7.

Table C in S1 Text) and to re-run the search with only *b*, *m*, *c* and *h* as unknown model parameters. The results (Fig 8) show that *b*, *m*, *c* and *h* are now well identifiable.

Of particular interest is that the posterior distribution obtained for *h* is very narrow. In the model, *h* quantifies the time scale of fluctuations of the individual cell responsiveness to light. This implies that learning *h* requires that single cells are tracked in time and that resulting time-correlation information is exploited for parameter inference. Algorithm 1 exploits this information by construction. Furthermore, *h* is a parameter of particular biological interest since it quantifies the fluctuations of a particular trait of cells (responsiveness to light), and hence how long the trait is inherited over cell generations. A priori, one would expect that a cell's responsiveness to light is determined by quantities such as plasmid copy numbers or the number of ribosomes and that these fluctuate on a time scale that is determined by the cells' growth rate. However, we find that the value for *h* that best explains our data is much smaller than what would be expected from the growth rate and that cells that responded more (or less) than average to light early in the experiment still tend to be more (or less) responsive several tens of cell generations later. The small value of *h* may suggest that variability in cell responsiveness to light is driven by a biological mechanism that generates atypically long memories

of the state of the cell. For instance, plasmid copy numbers might not fluctuate at the time scale of cellular growth due to feedback regulation that maintains high or low numbers. In particular, the ColE1 plasmid that carries a part of the CcaS/CcaR optogenetic system has been reported to multimerize, which might constitute a mechanism that creates large and long lasting fluctuations in its numbers.

## 4 Discussion

Despite immense advances in measurement technologies in the past decade, developing predictive models of natural or synthetically constructed gene networks remains a major challenge. Cell populations are heterogeneous while gene expression dynamics are stochastic and often regulated by intricate and poorly understood mechanisms. Understanding such processes in detail requires observations of sufficiently many single cells. Stochastic kinetic models are helpful to extract relevant information from the data and to test if a hypothesised biological mechanism can explain the observations. However, if years of modelling in systems biology have shown one thing, then it is that (some) parameters of mechanistic models of biochemical processes are almost always practically unidentifiable from the insufficiently informative data that is available [5]. This has led to increased efforts to quantify the information content of data and to develop methods to plan perturbations such that the studied system is excited in ways that reveal its dynamics [38, 39]. While well chosen perturbations may resolve some identifiability problems, it is to be expected that a single perturbation experiment will rarely be sufficient to ensure that the studied system displays all the dynamics that need to be observed to properly understand it. Novel experimental platforms that allow the user to parallelise different perturbations within a single experiment may resolve this problem but raise new questions and challenges.

We have focused here on microscopy platforms in which perturbations can be parallelised by targeting light signals at single cells. In principle, this allows one to test as many perturbations in parallel as there are cells in the experiment. Yet, in the face of stochasticity in the system and noise in the measurements, observing a few cells per perturbation implies that we will not be able to reliably observe the system's response to any of the employed perturbations. One might then wonder whether it is more informative to diversify input perturbations but to observe only few cells for each input or if one should rather ensure that many cells are observed for only few inputs. By studying parameter inference on simulated data for the model of the CcaS/CcaR optogenetic system, we show that exposing different cells to different perturbations within the same experiment (G0) provides means to increase the information content in the data and allows us to learn model parameters either more precisely or with fewer observed cells (Fig 5 and Fig D-F in S1 Text).

To reach these results, we had to deploy a method for the approximation of likelihoods that couples moment approximations, Gaussian assumptions, and a Kalman filter, since open loop likelihood evaluation would have been computationally infeasible for the considered data. Filtering-based methods may lead to vastly improved precision if the chemical master equation needs to be approximated. Relevant differences in approximation qualities appear when the measurements are taken frequently compared to the time scale of the system. Our results indicate that iterative and open loop likelihood calculation lead to the same approximation when the protein production rate is large in the case study of the positive feedback loop (Fig 2B), and that the quality of obtained parameter estimates becomes worse when measurement times are more distant for the studied model of a genetic toggle switch (Fig 3B and 3C).

Additionally, Algorithm 1 exploits the time-correlation information of every tracked cell, and enables us to learn some parameters of the model of the CcaS/CcaR optogenetic system from the experimental data, that are otherwise unidentifiable (Fig 8).

In this work we demonstrate that frequent measurements and the use of Algorithm 1 lead to very accurate likelihood approximations in all our case studies, even in cases where the used approximation of the CME is obviously bad. Importantly, increasing numbers of measurements increase the computational time of the approach only linearly (Fig 6) whereas open loop approaches that aim to calculate the joint distribution of all data points become essentially impossible to use when the number of measurement times ranges in the order of hundreds. With such data becoming more and more available, it can therefore be foreseen that parameter inference for stochastic kinetic models of biochemical reaction networks will necessarily have to be performed using iterative approaches in the future.

## 4.1 Experimental methods

**4.1.1 Bacterial strains and plasmids.** All experiments were performed with Escherichia coli strain HR14. The strain is derived by transformation of Escherichia coli MC4100 with three plasmids. Plasmids pSR43.6 and pSR58.6_3spng carry a CcaS/CcaR-based optogenetic module that drives or reduces expression from PcpcG2–172 promoters in green/red light respectively, and synthetic pathway for chromophore phycocyanobilin [40]. Plasmid pHyRep-Prg-0 carries a PcpcG2–172 regulated venus-YFP reporter, constitutive CFP reporter, and a stabilized repressilator without ssrA degradation tags [40]. pSR58.6_3spng also includes a repressor "sponge" region (tetR, lacI, and cI binding sites) from plasmid pLPT145 that reduces background repressor levels and reduces variability in repressilator period [41]. See table StrainsPlasmids.xlsx and plasmid maps PlasmidMaps.zip for composition in https://gitlab.pasteur.fr/adavidov/inferencelnakf. Cells were constructed and maintained in LB broth supplemented with Spectinomycin (100 $\mu$g/ml), Ampicillin (100 $\mu$g/ml), Chloramphenicol (20 $\mu$g/ml) as appropriate.

**4.1.2 Microscopic culture and optogenetic stimulation setup.** Experiments were performed as previously described, using an automated microscope platform for closed loop imaging and data processing, optogenetic stimulation, and environmental regulation [11].

Briefly, custom software operates an Olympus IX83 fluorescence microscope fitted with a 100x objective contained in an opaque, temperature-controlled incubator. The setup obtains CFP (x438/29,m483/22) and YFP (x513/22,m543/22) fluorescence images, derives cell size and expression reporter fluorescence levels, and delivers patterned optogenetic light stimuli to cells via a modified LCD projector with custom 530nm and 660nm light sources.

Bacterial cells are grown in microfluidic mother machines with 23$\mu$m×1.3$\mu$m×1.3$\mu$m (l,w, h) growth channels at 5$\mu$m spacing along a split media trench. The microfluidic devices are fabricated from degassed polydimethylsiloxane (Dow Sylgard 184, 1:10 catalyst:resin), cured against epoxy replicate master molds, ports punched, and plasma-bonded (Harrick PDC-002, medium power, 1 minute) to clean glass cover slips. See [11] for detailed protocol. Polyethylene tubing (Instech, BTPE-50) press-fitted to 22ga luer stubs and cannulae (Instech) is used to connect media and waste flows to the device. Media flow is regulated by two syringe pumps (WPI, Alladin-1000).

**4.1.3 Image acquisition, processing and cell stimulation loop.** The setup cycles through ten stage locations every six minutes, focusing and determining xy offsets, obtaining fluorescence images and delivering optogenetic stimuli to the cells [11]. To reduce optogenetic response to fluorescence imaging, exposures are minimized and maintained across experiments, and fluorescence image acquisition is immediately followed by application of light stimuli to set the cells' optogenetic state.

Briefly, at each timepoint and stage location, image-based autofocus and xy-stage jitter and drift corrections are first performed. Fluorescence images are then acquired and corrected for

small, slowly varying, additive camera signal offsets (by subtracting median dark images acquired alongside each), and shading corrected using previously obtained, normalized calibration images of a uniform fluorescent field (10% Fluorescein 0.1%NaHCO3) [42]. The images are spatially registered and fluorescence-based expression estimates for the constitutive and light-controlled reporters are extracted for individual mother cells as the 97th percentile pixel intensity within a pre-specified bounding box at each cell's image location. Constitutive reporter fluorescence images are segmented to derive cell sizes, and growth rates estimated via a moving average of differences in $\log_2$(cell length), excluding outliers due to cell division and segmentation errors.

The programmed CcaSR activation ($\sim$535nm) or deactivation ($\sim$670nm) light stimuli for each cell are then mapped to green or red boxes overlying the positions of the cells in an RGB image. The image is transformed to register projector to camera image planes, and projector shading corrections (using low-pass filtered reflected uniform field projections, obtained at experiment initialization) applied to each color channel. To stimulate the cells, this image is projected onto the field of view for ten seconds (670nm:$\sim$10.5 mW/cm$^2$, 535nm:7.6 mW/cm$^2$; Contrast relative to dark LCD panels, of 252 and 361, respectively; crosstalk between channels $<$1%).

**4.1.4 Experiment setup, media, and conditions.** Experiments were initialized and run as previously described [11]. Bacterial cells are diluted 1:100 from -80C glycerol stocks into 5ml fresh LB media containing 0.01% Tween20, 20$\mu$g/ml Chloramphenicol, 100$\mu$g/ml Spectinomycin, and 100$\mu$g/ml Ampicillin to maintain plasmids, and incubated for 6–7 hours at 37C. The experimental apparatus is equilibrated, and the microfluidic device prepared by filling with 0.01% Tween20 for 1 minute, then purging with air. The device is prewarmed, and cell culture concentrated (centrifuge 4000 $\times$g for 4 minutes, and resuspend pellet in 4 $\mu$l supernatant) and injected into the device. Media supply and waste tubes are fitted to the device and LB containing 0.4% glucose and 0.01% Tween20 delivered at 4 ml/hour for 1 hour, and 1.5 ml/hour—2.0 ml/hour thereafter.

With cells in the device, the experiment control software is initialized and calibrations (camera and projector offsets from the PDMS-glass interface, projector-camera transforms, and projector shading correction) performed. Fields of view are set, measurement regions for individual mother cells are specified and light sequences to be delivered are linked to each cell. $\sim$50 cells per field of view are distributed between seven repeating 100-pulse light sequences (consisting of initial green pulses of duration 0, 2, 5, 10, 20, 50, or 100 intervals followed by red pulses for the remaining intervals. All cells receive a common 100-pulse red "zeroing" sequence during the first 10 hours of the experiment. The system then continues to acquire data and stimulates cells according to their assigned light sequences for the remainder of the experiment. For the model calibration the full red sequence is not relevant and is left out and the common initial 100-pulse red "zeroing" sequence is shortened to the duration of 4 intervals. Data with no pulse (0 pulse duration) is not used for the model calibration as it is a trivial case and no parameter could be learnt with it.

**4.1.5 Cell classification and invalidation.** Cells that stop growing or filament in the microfluidic device, or that lose plasmids or optogenetic system function, or transiently shift from the detection/excitation regions can give unreliable data. Our automated setup therefore continuously tests and permanently flags cells that fail presence, growth, and optogenetic system function tests [11]. Briefly, the constitutively-expressed reporter (here, CFP) is used to test control location presence and measurement quality (invalidating cells for signal loss or noise above threshold). The constitutive reporter images are also used for cell shape and growth rate extraction (invalidation for growth below threshold rate), and pSR43.6 loss detection (invalidation for resulting growth rate increase and reduced CFP concentration). Optogenetic response

is also measured via the linked reporter (YFP) and cells can be flagged for response below minimum threshold (indicating pSR58.6_3spng or optogenetic system loss). Cells failing individual tests are typically classified as invalid for the remainder of the experiment. As the automated classifier may fail to invalidate pathological cells, we followed the standard invalidation protocol above with a manual cull of remaining cells that presented improbable and outlier trajectories to our eye, for example, cells that never react to the light signal or those that after some time show a non-zero flat behaviour, that we assume to be dead.

## Supporting information

**S1 Text. Supporting information.**
(PDF)

## Acknowledgments

We thank Virgile Andreani for useful discussions about the model and parameter inference. We thank Johan Paulsson and Jeffrey J Tabor for kind gifts of plasmids.

## Author Contributions

**Conceptualization:** Jakob Ruess.

**Data curation:** Anđela Davidović, Remy Chait.

**Formal analysis:** Anđela Davidović, Jakob Ruess.

**Investigation:** Anđela Davidović, Jakob Ruess.

**Methodology:** Jakob Ruess.

**Project administration:** Jakob Ruess.

**Resources:** Remy Chait.

**Software:** Anđela Davidović.

**Supervision:** Gregory Batt, Jakob Ruess.

**Validation:** Anđela Davidović.

**Visualization:** Anđela Davidović.

**Writing – original draft:** Anđela Davidović, Remy Chait, Gregory Batt, Jakob Ruess.

**Writing – review & editing:** Anđela Davidović, Remy Chait, Gregory Batt, Jakob Ruess.

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
