## [Decision Letter · Decision Letter 0]

14 Oct 2021

Dear Dr. Ruess,

Thank you very much for submitting your manuscript "Parameter inference for stochastic biochemical models from perturbation experiments parallelised at the single cell level" for consideration at PLOS Computational Biology.

As with all papers reviewed by the journal, your manuscript was reviewed by members of the editorial board and by several independent reviewers. In light of the reviews (below this email), we would like to invite the resubmission of a significantly-revised version that takes into account the reviewers' comments.

We cannot make any decision about publication until we have seen the revised manuscript and your response to the reviewers' comments. Your revised manuscript is also likely to be sent to reviewers for further evaluation.

Sincerely,

Christopher Rao

Associate Editor

PLOS Computational Biology

Jason Haugh

Deputy Editor

PLOS Computational Biology

Reviewer's Responses to Questions

**Comments to the Authors:**

Reviewer #1: The paper by Davidovic et al proposes a method for performing inference of parameters of stochastic reaction networks from time series data. The paper contains a theoretical description of the algorithm as well as extensive validation on simulated data and on one new experimental data set. Overall, the paper is well written and very clear, although cosmetic improvements to the notation could be made. My major reservation though is that I do not understand why the methodology should be considered new. As far as I know, the idea of performing Kalman filtering with an approximation of the transition probabilities for stochastic reaction networks has been initially proposed in an overlooked paper by Ruttor and Opper (Phys. Rev. Lett. 103, 2009) over ten years ago. Several others have followed suit. It is true that several papers use the LNA or similar in an open loop approach, but it is well known that this is simply wrong (see e.g. the popular review by Schnoerr et al in J.Phys.A 2017, sec 6.2-6.4, in particular the incipit of 6.4.1 which very clearly explains how the problem should be formulated in terms of approximating the transition probability in a forward-backward algorithm, which is what the authors of this paper do). Having said that, many of the empirical results of this paper are interesting, e.g. the study of how sampling time affect accuracy, and also and perhaps primarily the real data study. I think the authors should: either be much clearer about the novelty of their method (in case my diagnosis above is wrong), or completely restructure the paper, removing claims of algorithmic novelty and focussing on the analysis and the data part.

Reviewer #2: The manuscript presents an interesting extension of the moment based methods for inferences of parameters governing molecular processes inside single cells. It offers to bypass disadvantages related to a fixed form of transition probabilities by coupling inference to a Kalman filter that is updated over time. The applicability and advantages of the approach are well documented using synthetic and experimental data (opto-genetic gene activation system in E. coli). The text is well written and easy to follow. My critique should include to major points:

From my understanding, the different responses of single-cells in section 3.2.3 are assumed to result from noise that is modelled by LNA and Kallman Filter. This is however not necessarily (most likely in my view) the case. Most likely the differences result from differences in copy number of molecular species involved in the system (often referred to as extrinsic noise). If my understanding is correct then treating the data like this is misleading. Several approaches can be easily found that model extrinsic variability.

I am not sure how novel the ideas to couple Kallman Filter into LNA/Moment based inference is. This should be explained in more detail in the introduction.

Reviewer #3: Please see the attached.

**Have the authors made all data and (if applicable) computational code underlying the findings in their manuscript fully available?**

Reviewer #1: None

Reviewer #2: Yes

Reviewer #3: Yes

PLOS authors have the option to publish the peer review history of their article (what does this mean?). If published, this will include your full peer review and any attached files.

Reviewer #1: No

Reviewer #2: No

Reviewer #3: No
---

## [Decision Letter · Decision Letter 1]

13 Feb 2022

Dear Dr. Ruess,

Thank you very much for submitting your manuscript "Parameter inference for stochastic biochemical models from perturbation experiments parallelised at the single cell level" for consideration at PLOS Computational Biology. As with all papers reviewed by the journal, your manuscript was reviewed by members of the editorial board and by several independent reviewers. The reviewers appreciated the attention to an important topic.

All three reviewers recommended publication. However, reviewer 3 offered two minor suggestions for improving the manuscript.  We ask that you consider these recommendation before we accept that final version of the manuscript. 

Sincerely,

Christopher Rao

Associate Editor

PLOS Computational Biology

Jason Haugh

Deputy Editor

PLOS Computational Biology

[LINK]

Reviewer's Responses to Questions

**Comments to the Authors:**

Reviewer #1: The authors have done a good job in repositioning the paper to highlight the more novel and interesting aspects. I'm happy for it to proceed as is

Reviewer #2: All concerns were addressed to my satisfaction.

Reviewer #3: The authors have carefully revised the manuscript to address my concerns regarding the previous version. Most importantly, the revision has emphasized more clearly the contribution of the paper, which is to show how new experimental platforms that produce multiple single-cell trajectories measured at high frequencies may enable more precise and efficient identification of model parameters, and how methods based on LNA and moment closure, which are notoriously inaccurate for other types of data, may become serious contenders for the new data types. In my opinion, this is an interesting insight that deserves publication.

While the current version is acceptable, I have minor suggestions that in my opinion could help improve the clarity of writing.

Minor comments

1. It appears to me the current title of the manuscript still seems to put the reader’s focus more on the “parameter inference” part rather than the new experimental data type and empirical results that the authors aim to emphasize. For example, how about “New perturbation experiments parallelized at the single cell level enables efficient parameter inference for stochastic biochemical models”?

2. On line 601, the authors claimed that "it has not been recognised that iterative likelihood evaluations may lead to vastly improved precision if the chemical master equation needs to be approximated". I have reservation about this claim, since it seems in much of literature on parameter inference of stochastic models that iterative approaches (such as particle filtering, particle marginal MCMC...) are the methods of choice when dealing with time-series data. The main question is what simulation/approximation of the CME (SSA, FSP, LNA...) is effective and efficient on the data at hand. Admittedly, when the data were sparse with long time intervals between measurements, many methods became either very slow or imprecise.

**Have the authors made all data and (if applicable) computational code underlying the findings in their manuscript fully available?**

Reviewer #1: None

Reviewer #2: Yes

Reviewer #3: Yes

PLOS authors have the option to publish the peer review history of their article (what does this mean?). If published, this will include your full peer review and any attached files.

Reviewer #1: No

Reviewer #2: No

Reviewer #3: No

Figure Files:

Data Requirements:

Reproducibility:

References:

---

## [Editor Report · Decision Letter 2]

21 Feb 2022

Dear Dr. Ruess,

We are pleased to inform you that your manuscript 'Parameter inference for stochastic biochemical models from perturbation experiments parallelised at the single cell level' has been provisionally accepted for publication in PLOS Computational Biology.

Best regards,

Christopher Rao

Associate Editor

PLOS Computational Biology

Jason Haugh

Deputy Editor

PLOS Computational Biology

---

## [Editor Report · Acceptance letter]

14 Mar 2022

PCOMPBIOL-D-21-01585R2 

Parameter inference for stochastic biochemical models from perturbation experiments parallelised at the single cell level

Dear Dr Ruess,

I am pleased to inform you that your manuscript has been formally accepted for publication in PLOS Computational Biology. Your manuscript is now with our production department and you will be notified of the publication date in due course.

With kind regards,

Zsanett Szabo
